# A Fast Point Clouds Registration Algorithm Based on ISS-USC Feature for the 3D Laser Scanner

**Aihua Wu, Yinjia Ding, Jingfeng Mao * and Xudong Zhang**

School of Mechanical Engineering, Nantong University, Nantong 226019, China
* Correspondence: mao.jf@ntu.edu.cn

**Abstract:** The point clouds registration is a key step in data processing for the 3D laser scanner to obtain complete information of the object surface, and there are many algorithms. In order to overcome the disadvantages of slow calculation speed and low accuracy of existing point clouds registration algorithms, a fast point clouds registration algorithm based on the improved voxel filter and ISS-USC feature is proposed. Firstly, the improved voxel filter is used for down-sampling to reduce the size of the original point clouds data. Secondly, the intrinsic shape signature (ISS) feature point detection algorithm is used to extra feature points from the down-sampled point clouds data, and then the unique shape context (USC) descriptor is calculated to describe the extracted feature points. Next, the improved random sampling consensus (RANSAC) algorithm is used for coarse registration to obtain the initial position. Finally, the iterative closest point (ICP) algorithm based on KD tree is used for fine registration, which realizes the transform from the point clouds scanned by the 3D laser scanner at different angles to the same coordinate system. Through comparing with other algorithms and the registration experiment of the VGA connector for monitor, the experimental results verify the effectiveness and feasibility of the proposed algorithm, and it has fastest registration speed while maintaining high registration accuracy.

**Keywords:** point clouds registration; voxel filter; intrinsic shape signatures; unique shape context; random sampling consensus; iterative closest point



## 1. Introduction

With the rapid development of 3D laser scanning technology, it has been widely used in many fields such as robotics [1], reverse engineering [2], geological survey [3] and cultural protection [4]. Due to the influence of the angle of the scanning device, the shape of the scanned object and environmental factors, it is impossible to complete the data collection of the physical scanning at one time. In order to obtain the complete 3D information of the object surface, it is necessary to collect data from multiple angles and blocks of the object, and then splice or register the 3D point clouds collected from different angles, so as to obtain the point clouds containing complete information of the object under the same coordinate system.

The point clouds registration [5] is a key step to obtain complete information of the object surface, and it also plays an important role in 3D reconstruction, 3D localization and pose estimation. Take the palletizing and sorting robot as an example, which have been widely used in food, medicine, chemical and other automatic production enterprises, point clouds registration technology is an essential step. It can also provide high-precision services for intelligent mobile robots.

At present, the most widely used and classic registration algorithm is the iterative closest point (ICP) algorithm [6]. The algorithm is simple, but it requires a good initial position, and two point clouds must have overlapping parts, otherwise it is easy to fall into the local optimal solution, which leads to poor final registration effect. In recent years, researchers have put forward many improvement schemes based on the original

ICP algorithm. For example, reference [7] proposes a cluster iterative closest point method named CICP for sparse-dense point clouds registration, which is a new method that surpassed the concept of density. It can handle registration of point clouds of different densities acquired by the same sensor at different resolutions or from different sensors. Reference [8] proposes a global optimal algorithm named Go-ICP, which integrates the local ICP into the BnB scheme. It can not only ensure the global optimality but also improve the speed of the algorithm. Reference [9] proposes LieTrICP algorithm. It is a robust registration method of two-point sets based on Lie group parameterization, which combines the advantages of Trimmed Iterative Closest Point (TrICP) and Lie group representation, making the algorithm more robust and accurate. Some researchers first use the coarse registration to obtain a good initial registration position, and then use ICP algorithm for fine registration. Reference [10] proposes a feature descriptor based on the rotation volume ratio to describe feature points. Based on the feature point descriptor, coarse registration is used to obtain a good initial transformation matrix, and then the improved ICP algorithm is used to obtain precise transformation matrix. Reference [11] proposes the Fast Point Feature Histogram (FPFH) descriptor, and then the Best-Bin-First (BBF) is used to reduce the data dimension, which greatly accelerates the iteration speed of ICP. Reference [12] first uses sample consensus initial alignment algorithm (SAC-IA) for coarse registration, and then iterative closest point algorithm based on point-to-face is used for fine registration. Other researchers have proposed point clouds registration algorithms different from the ICP model, such as the three-dimensional normal distributions transform (3D-NDT) algorithm based on the probability density model [13], which does not need to calculate the nearest neighbor matching points, and thus improves the calculation efficiency. The 4PCS (4-points congruent sets) algorithm [14] reduces spatial matching operations by constructing and matching congruent four-point pairs, which accelerates the registration process. Reference [15] uses the deep learning model for point clouds registration. It proposes a new unsupervised deep learning network-Binary Tree Network (BTreeNet), which learns the features of rotation and translation matrices, respectively.

Point clouds registration methods can be divided into two categories: optimization-based methods, and feature-based methods. Optimization-based methods search for corresponding point pairs in the source point clouds and the target point clouds, and the transformation matrix is estimated according to the correspondence. Above two steps will be iterated to acquire the best transformation matrix. The disadvantage of this kind of methods is that complex strategies are required to suppress noise, outliers and density changes, which can increase computational burden. Feature-based methods do not search for corresponding point pairs, they extract the feature points from the source point clouds and the target point clouds. Then the feature descriptor is used to describe them, and the feature is used to estimate the transformation matrix. Therefore, the selection of feature points and feature descriptor must affect the registration effect. Due to the lack of representativeness or insufficient number of feature points, these methods tend to have low registration accuracy.

By analyzing and comparing the current point clouds registration algorithms, in order to improve the registration speed and accuracy, this paper proposes a point clouds registration algorithm based on improved voxel filter combining ISS feature points detection [16] algorithm and the USC [17] descriptor. The ISS feature points detection algorithm is used to extract feature points from the point clouds with the improved voxel filter, and the extracted feature points are described by the USC descriptor. Based on the initial registration position obtained after the improved RANSAC coarse registration [18], the ICP algorithm based on KD tree [19] is used to complete the final point clouds fine registration.

In summary, the main contributions of this paper include: (1) The traditional voxel filter is improved, and the point closest to the voxel center of gravity in the original point clouds data is used to replace the voxel center of gravity. This method not only maintains the tiny features of the original point clouds, but also improves the accuracy of the point clouds data, which is more conducive to the description of the surface corresponding to the

sampling point. (2) USC feature descriptors are used to describe the extracted key points, it only needs to calculate a certain descriptor on each feature point instead of multiple fuzzy descriptors, which reduces memory usage and improves computational efficiency. (3) In the process of RANSAC coarse registration, adding a pre-exclusion step can immediately filter out the wrong hypothetical poses, thereby saving more time to generate more other possibly correct hypothetical poses and reducing the time of coarse registration. (4) The feasibility and effectiveness of the proposed algorithm are verified by comparing with other algorithms and the registration experiment of the VGA connector for monitor.

The rest of this paper is organized as follows. Section 2 briefly introduces the proposed algorithm flow. Section 3 introduces the principles of the five main steps of the proposed algorithm in detail. Section 4 discusses the comparison experimental results of the proposed algorithm and other algorithms on several models, and the registration experiment of the VGA connector for monitor, as well as evaluates their accuracy and effectiveness. Finally, Section 5 is the conclusion of this paper.

## 2. Algorithm Principle

The basic principle of the proposed algorithm mainly includes five processes. Firstly, an improved voxel filter is used to down sample the source point clouds and the target point clouds, respectively. Secondly, the feature points are extracted by the ISS feature points detection algorithm to obtain a set of points with rich geometric feature information. Then the extracted feature points are described with the USC descriptor to form ISS-USC feature points descriptor. Next, the improved RANSAC algorithm is used for coarse registration to obtain the optimal initial transformation matrix. Finally, the ICP algorithm based on KD tree is used for fine registration. According to the final obtained transformation matrix, the source point clouds are transformed into the coordinate system under the target point clouds to complete the registration. The algorithm flow is shown in Figure 1.

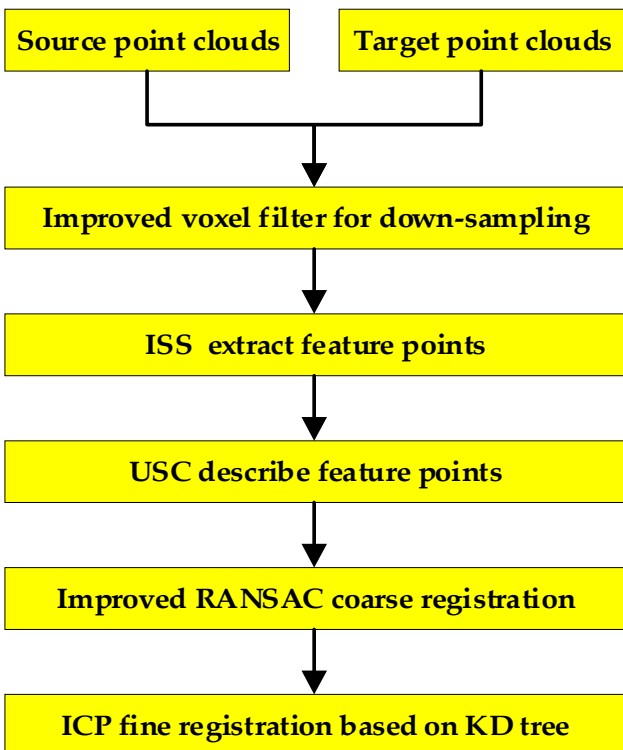

**Figure 1.** Algorithm flow chart.

### 3. Algorithm Design

*3.1. Improved Voxel Filter for Down-Sampling*

Voxel filter [20] is to create a 3D voxel grid for the input point clouds data. It uses the center of gravity of all points in each voxel to approximate display other points in the voxel, that is, uses the center of gravity point to represent all points within this voxel. Using voxel filter can not only reduce the number of points, but also maintain the shape characteristics of the point clouds. However, when traditional voxel filter is used, the center of gravity point is not necessarily a point in the original point clouds, which will lose the fine features of the original point clouds. Therefore, it is necessary to improve the voxel filter. K-nearest neighbor search [21] is performed on the center of gravity points with voxel filter of the point clouds. For a given query point, that is, the center of gravity point, search its nearest neighbor, and K = 1 at this time. The point closest to the voxel center of gravity in the original point clouds data is used to replace the voxel center of gravity, and then perform the same processing on all voxels to acquire the filtered point clouds. The improved voxel filter not only reduces the execution time of the algorithm, but also effectively improves the accuracy of the data.

Figure 2a is the Bunny model of the Stanford University point clouds library, which has more than 30,000 points. In order to reduce the execution time of the algorithm, the improved voxel filter is used to simplify the point clouds. Figure 2b shows the results of the down-sampling using the improved voxel filter. After down-sampling, the number of point clouds is significantly reduced, only more than 3000 points, and the shape characteristics of the point clouds remain unchanged.

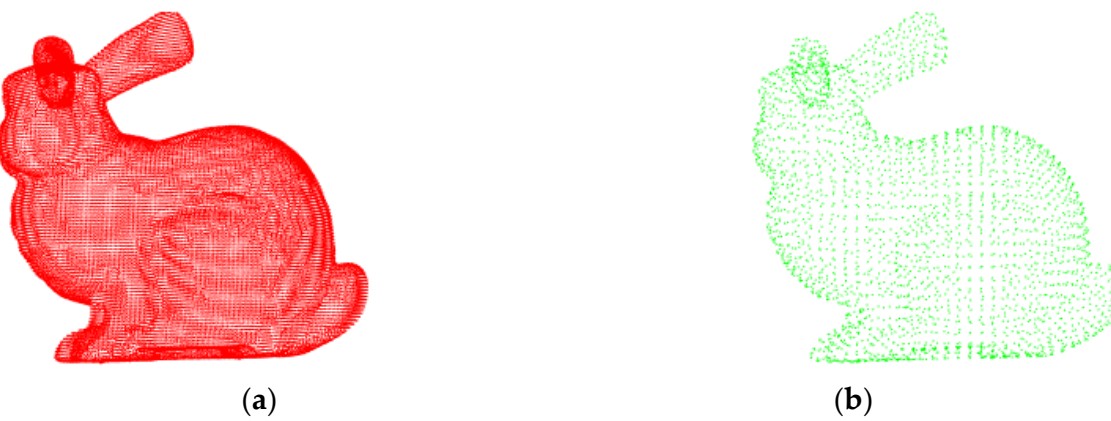

(**a**)          (**b**)

**Figure 2.** Bunny model and down-sampling results. (**a**) Bunny model, (**b**) down-sampling results.

Figure 3 shows the original point clouds, the point clouds after voxel filter and the point clouds after improved voxel filter of the Bunny model. The original point clouds are set to green, the point clouds after voxel filter are set to blue, and the point clouds after improved voxel filter are set to red. Three point clouds are put together for comparison. The red point clouds and the green point clouds actually coincide. The blue point clouds are the center of gravity of each voxel, which are different from the original point clouds. Therefore, the improved voxel filter will not change the small features of the original point clouds even after sampling, which can improve the point cloud data expression accuracy.

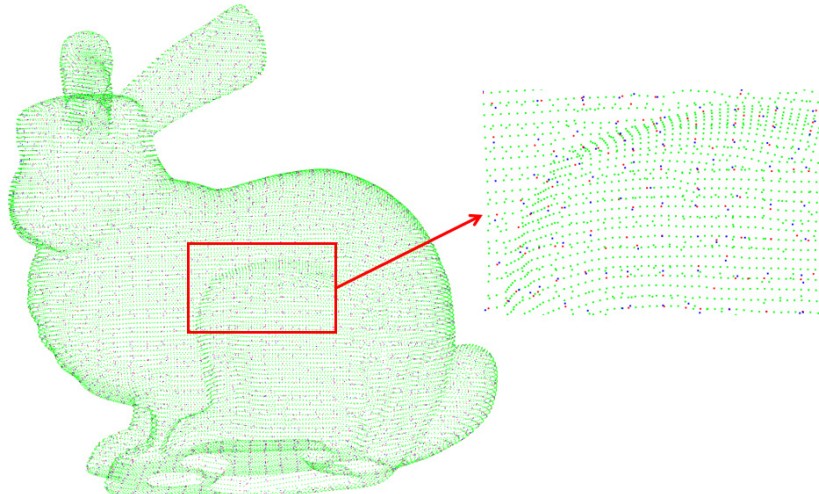

**Figure 3.** The original point clouds, the point clouds after voxel filter and the point clouds after improved voxel filter of the Bunny model.

*3.2. ISS Feature Points Detection Algorithm*

The intrinsic shape signatures (ISS) feature points detection algorithm is based on eigenvalue decomposition of covariance matrix, which has rich geometric feature information. The main steps of extracting ISS feature points of 3D point clouds data $P$ are as follows:

(1) Set a search radius $r_{ISS}$ for each point $P_i$ in $P$, and calculate the weight $w_{ij}$ of all points in the area with $P_i$ as the center and $r_{ISS}$ as the radius, as shown in Formula (1):

$$w_{ij} = \frac{1}{|p_i - p_j|}, |p_i - p_j| < r_{ISS} \tag{1}$$

where $P_j$ is any point within the area with $P_i$ as the center and $r_{ISS}$ as the radius.

(2) Calculate the covariance matrix $cov(P_i)$ of each point $P_i$, as shown in Formula (2):

$$\text{cov}(p_i) = \frac{\sum\limits_{|p_i-p_j|<r_{ISS}} w_{ij}(p_i - p_j)(p_i - p_j)^T}{\sum\limits_{|p_i-p_j|<r_{ISS}} w_{ij}} \tag{2}$$

(3) Calculate the eigenvalues $\{\lambda_i{}^1, \lambda_i{}^2, \lambda_i{}^3\}$ of the covariance matrix $cov(P_i)$, and arrange $\lambda_i{}^1, \lambda_i{}^2, \lambda_i{}^3$ in decreasing order of size, and regard the feature points satisfying the Formula (3) as candidate feature points of intrinsic shape signatures:

$$\frac{\lambda_i^2}{\lambda_i^1} \le \varepsilon_1, \frac{\lambda_i^3}{\lambda_i^2} \le \varepsilon_2 \tag{3}$$

where $\varepsilon_1$ and $\varepsilon_2$ are the set thresholds, $0 < \varepsilon_1, \varepsilon_2 < 1$;

(4) For the candidate feature points of intrinsic shape signatures, the non-maximum suppression strategy is used to filter out the final feature points.

*3.3. USC Descriptor*

Commonly, feature point description methods include Spin Image descriptor [22], PFH descriptor, FPFH descriptor [23] and 3DSC descriptor. Reference [24] proposes PFH feature descriptor and FPFH feature descriptor, and uses FPFH feature descriptor and SAC-IA (SAmple Consensus Initial Alignment) for registration. Reference [25] proposes a new 3D descriptor LP-PPF, which can identify repetitive structures correctly and achieve accurate registration between adjacent point clouds pairs.

3D Shape Context [26] (3SDC) is a 3D local feature point descriptor extended from 2D shape context. Reference [27] uses the 3DSC descriptor to describe the extracted feature points, then RANSAC algorithm is used for coarse registration and ICP algorithm is used for fine registration. Compared with the ICP algorithm, the SAC-IA + ICP algorithm and the 3DHoPD + ICP algorithm, the 3DSC + RANSAC + ICP has a faster registration speed. Figure 3a shows the meshing diagram of 3DSC. It takes the normal vector *n* of a point as the local reference coordinate axis, and its disadvantage is that it lacks a repeatable local reference coordinate system and needs to calculate descriptors in multiple rotations in different directions. Therefore, the 3DSC descriptor can be improved to use the unique shape context (USC) descriptor. It only needs to calculate one descriptor at each model feature, which not only reduces the memory consumption, but also improves the calculation efficiency. The main steps for USC to describe feature points are as follows:

(1) Given a feature point *k* and a spherical neighborhood with *k* as the center and *R* as the radius, the weighted covariance matrix *M* of the points in the neighborhood is shown in Formula (4):

$$
\begin{cases}
M = \frac{1}{Z} \sum\limits_{s:d_s \leq R} (R - d_s)(k_s - k)(k_s - k)^T \\
d_s = \|k_s - k\| \\
Z = \sum\limits_{s:d_s \leq R} (R - d_s)
\end{cases}
\tag{4}
$$

where $k_s$ is any point in the spherical neighborhood, and *s* is the number that satisfies the condition of $\|k_s - k\| \leq R$.

(2) According to the eigenvector decomposition of *M*, the local reference coordinate system of the feature point is determined, and the spherical neighborhood around the feature point is uniquely divided into grids along the three coordinate axes of the local reference coordinate system. The weighted value of the points in each grid is the USC descriptor, as shown in Formula (5):

$$
w(k_j) = \frac{1}{\rho_j \sqrt[3]{V(x,y,z)}}
\tag{5}
$$

where $V(x,y,z)$ represents the volume of *x* direction, *y* direction and *z* direction, and $\rho_j$ is the density of points in the corresponding volume.

Figure 4 shows the meshing diagram of 3DSC and USC.

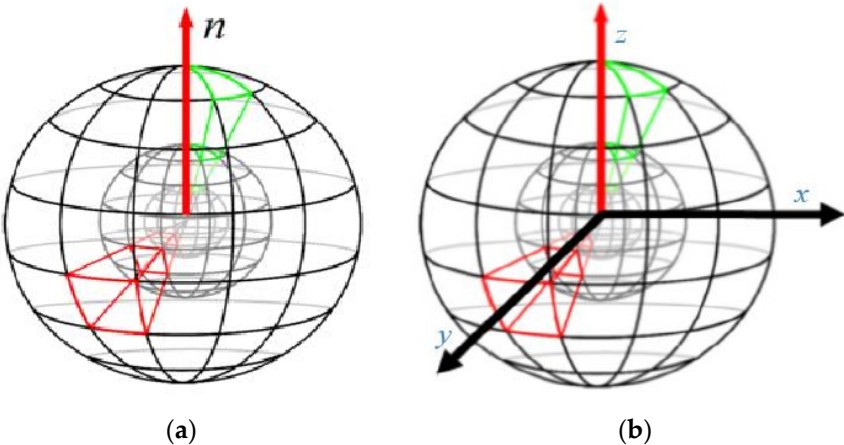

(a)　　　　　　　　　　　　　　　　　(b)

**Figure 4.** The meshing diagram of 3DSC and USC. (**a**) The meshing diagram of 3DSC. (**b**) The meshing diagram of USC.

### 3.4. Improved RANSAC Coarse Registration

The goal of point clouds registration is to estimate the transformation matrix so that the sum of squared distances between each point $p$ in the source point clouds $P$ and the corresponding point $q$ in the target point clouds $Q$ is minimized, as shown in Formula (6):

$$\hat{T} = \underset{T}{\mathrm{argmin}}\,\varepsilon(T) = \underset{T}{\mathrm{argmin}} \sum_{p \in P} (T_p - q)^2 \tag{6}$$

The main steps of the coarse registration of the random sample consensus algorithm are as follows:

(1) Select $n$ random sample points $p_1, p_2, \ldots, p_n$ in the source point clouds $P$, and search for $n$ corresponding points $q_1, q_2, \ldots, q_n$ in the target point clouds $Q$ through nearest neighbor matching according to the USC descriptor, so that $n$ corresponding point pairs $(p_1, q_1), (p_2, q_2), \ldots, (p_n, q_n)$ can be obtained, where $n \geq 3$;

(2) The pre-exclusion step to avoid incorrect hypothetical poses. First, calculate the Euclidean distance between $n$ points in their respective spaces, and $n$ random sample points will form the side lengths of multiple virtual polygons. Second, the dissimilar vector $\vec{\delta}$ between the side lengths of virtual polygons is calculated. Finally, compare $\vec{\delta}$ with the preset side length similarity threshold $\varepsilon_{ploy}$. If $\|\vec{\delta}\| \leq \varepsilon_{ploy}$, continue to the next step, otherwise return to the first step. Taking $n = 3$ as an example, the calculation formula $\vec{\delta}$ is shown in Formula (7):

$$\overline{\delta} = \left[ \frac{d_{12}^p - d_{12}^q}{\max(d_{12}^p, d_{12}^q)}, \frac{d_{23}^p - d_{23}^q}{\max(d_{23}^p, d_{23}^q)}, \frac{d_{13}^p - d_{13}^q}{\max(d_{13}^p, d_{13}^q)} \right] \tag{7}$$

The calculation principle of $\vec{\delta}$ is shown in Figure 5:

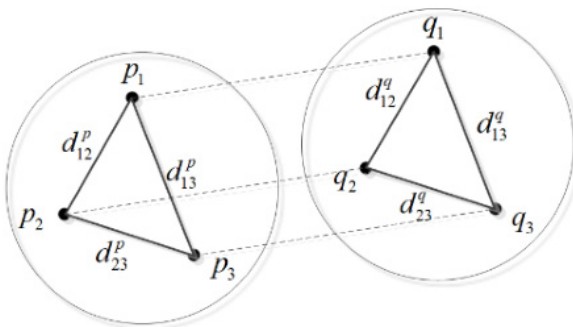

**Figure 5.** The calculation principal diagram of $\vec{\delta}$.

(3) Use $n$ corresponding point pairs $(p_1, q_1), (p_2, q_2), \ldots, (p_n, q_n)$ to estimate the transformation matrix $\hat{T}$, and the source point clouds $P$ is transformed into point clouds $P'$ after $\hat{T}$ transformation. The nearest neighbor Euclidean distance between the point clouds $P'$ and the target point clouds $Q$ is then calculated. Next, let the point which its nearest Euclidean distance in the point clouds $P'$ is less than the preset distance threshold $\varepsilon_{RANSAC}$ as the interior point, and calculate the distance between the interior point and the corresponding point by the formula of $\varepsilon(T) = \sum_{p' \in P'} (p' - q)^2$. If the number of interior points is too small, return to the first step;

(4) Re-estimate the transformation matrix $\hat{T}$ according to the relationship between the interior point and the corresponding point in the target point clouds $Q$, and perform continuous iteration. When $\varepsilon(T)$ reaches the minimum value or reaches the maximum number

of iterations $t_{RANSAC}$, the iteration is stopped, and the optimal transformation matrix $\hat{T}_0$ is obtained at this time. The calculation formula of $t_{RANSAC}$ is shown in Formula (8):

$$t_{RANSAC} = \frac{\log(1-\rho)}{\log(1-\omega^n)} \tag{8}$$

where $\rho$ is the given expected success probability and $\omega$ is the expected inline score. The expected inline score is the percentage of interior points in a data set that contains both interior points (points that are suitable for the model) and exterior points (points that are not suitable for the model).

### 3.5. ICP Fine Registration Based on KD Tree

The iterative closest point algorithm is the most classic point clouds registration algorithm, but it is easy to fall into the local optimal solution. Therefore, it is necessary to provide a good initial position through coarse registration to increase the probability of iterative convergence to the global optimal position. ICP fine registration based on KD tree includes the following steps:

(1) Perform RANSAC coarse registration on the source point clouds $P$ to obtain the optimal transformation matrix $\hat{T}_0$, and perform $\hat{T}_0$ transform on the point clouds $P$ to obtain the point clouds $P''$. For each point $p_i''$ in the point clouds $P''$, use the KD tree algorithm to search for the point $q_i$ closest to the point $p_i''$ in the target point clouds $Q$, and form a corresponding point pair $(p_i'', q_i)$. This makes up a total of $N$ pairs of points;

(2) According to the point pair relationship, the rotation matrix $R$ and the translation matrix $T$ are calculated by the least square method, and the error function $G(R,T)$ is obtained. The calculation formula is shown in Formula (9):

$$G(R, T) = \frac{1}{N} \sum_{i=1}^{N} \left\| R_{p_i''} + T - q_i \right\|^2 \tag{9}$$

(3) Transform each point $p_i''$ in the point clouds $P$ with the obtained rotation matrix $R$ and translation matrix $T$ to obtain the transformed corresponding point $p'''$, $p_i''' = R_{p_i''} + T$, and calculate the average distance $D$ of the point pair $(p''', q_i)$, as shown in Formula (10):

$$D = \frac{1}{N} \sum_{i=1}^{N} \left\| p_i''' - q_i \right\| \tag{10}$$

(4) Set the maximum distance threshold $\varepsilon_{ICP}$ between corresponding points. If $D_m - D_{m+1} \leq \varepsilon_{ICP}$ is satisfied, the iteration ends. $D_m$ is the average distance of the $m$th iteration, and $D_{m+1}$ is the average distance of the $m+1$th iteration. Otherwise, return to the first step to change the optimal transformation matrix and continue the iteration, at this time the initial transformation matrix is $T_0 \times R \times T$.

(5) According to the final obtained rotation matrix $R$ and translation matrix $T$, the source point clouds $P$ is transformed into the coordinate system of the target point clouds $Q$ to complete the registration.

## 4. Experimental Results and Analysis

### 4.1. Experimental Data

The experimental data in this paper adopts the Bunny, Armadillo, Dragon and Drill models in the point clouds database of Stanford University. The original scale of Bunny1 and Bunny2 point clouds are 35,974, and the original scale of Armadillo1 and Armadillo2 point clouds are 204,800. The original scale of Dragon1 and Dragon2 point clouds are 29,103, and the original scale of Drill1 and Drill2 point clouds are 204,800. The experimental hardware CPU is Intel(R) Core(TM) i5-9400 @ 2.90 GHz processor, the operating system

software is Windows 10 Enterprise Edition, and the development environment is PCL 1.8.0 and visual studio 2013.

In order to reduce the execution time of the algorithm, the improved voxel filter is first used for down-sampling to simplify the point clouds, and control the scale of points after down-sampling to several thousand. Too many feature points will increase the execution time of the algorithm, and too few feature points will affect the registration accuracy.

Table 1 shows the number of points and ISS feature points of the Bunny model under different grid sizes. The grid size of the Bunny point clouds data is selected by experiment to be 0.005 m, and the grid size of the Armadillo point clouds data is 0.003 m. The grid size of the Dragon point clouds data is selected by experiment to be 0.03 m, and the grid size of the Drill point clouds data is 0.001 m.

**Table 1.** The number of points and feature points of Bunny model under different grid sizes.

| Grid Size/m | 0.001 | 0.002 | 0.003 | 0.004 | 0.005 | 0.006 | 0.007 |
|---|---|---|---|---|---|---|---|
| Bunny1 points | 34,461 | 15,897 | 7822 | 4639 | 3006 | 2131 | 1567 |
| ISS feature points 1 | 66 | 1020 | 425 | 245 | 158 | 105 | 74 |
| Bunny2 points | 33,940 | 16,243 | 8053 | 4771 | 3112 | 2188 | 1605 |
| ISS feature points 2 | 59 | 1048 | 460 | 255 | 162 | 107 | 73 |

The neighborhood radius searched by the algorithm and grid size are the same, which are 0.005 m, 0.003 m, 0.03 m and 0.001 m, respectively. The final feature points are filtered out by non-maximum suppression strategy, and the selected non-maximum suppression radius is twice the radius of the searched neighborhood, which is 0.01 m, 0.006 m, 0.06 m and 0.002 m, respectively. The final scale of improved voxel filter down-sampling points and extracted feature points are shown in Table 2. Figures 6 and 7 are the feature point plots extracted from Bunny1, Bunny2, ArmAdillo1 and ArmAdillo2 models, respectively, after down-sampling with improved voxel filter. Figures 8 and 9 are the feature point plots extracted from Dragon1, Dragon2, Drill1 and Drill2 models, respectively, after down-sampling with improved voxel filter.

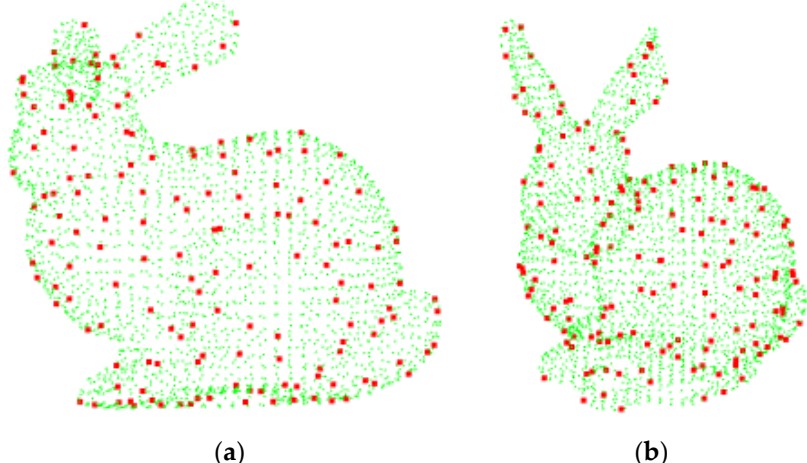

(**a**)　　　　　　　　　　　(**b**)

**Figure 6.** Feature points plot extracted from Bunny1 and Bunny2 models. (**a**) Bunny1 model. (**b**) Bunny2 model.

**Table 2.** The number of point clouds for different models.

| Model | Original Point Clouds | Improved Voxel Filter Down-Sampling Points | ISS Feature Points |
|---|---|---|---|
| Bunny1 | 35,947 | 3006 | 158 |
| Bunny2 | 35,947 | 3112 | 162 |
| Armadillo1 | 204,800 | 2822 | 147 |
| Armadillo2 | 204,800 | 2758 | 138 |
| Dragon1 | 29,103 | 5611 | 283 |
| Dragon2 | 29,103 | 5704 | 272 |
| Drill1 | 204,800 | 1551 | 108 |
| Drill2 | 204,800 | 1503 | 103 |

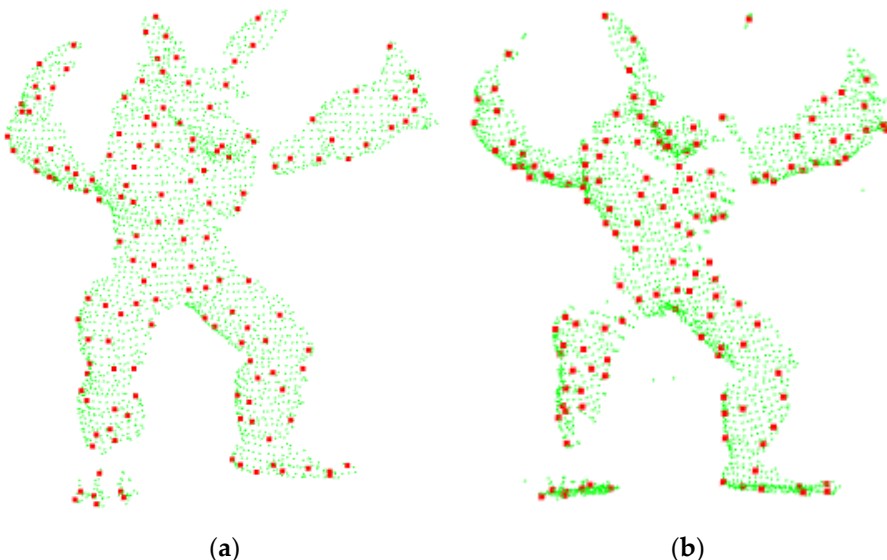

(**a**)        (**b**)

**Figure 7.** Feature points plot extracted from ArmAdillo1 and ArmAdillo2 models. (**a**) ArmAdillo1 model. (**b**) ArmAdillo2 model.

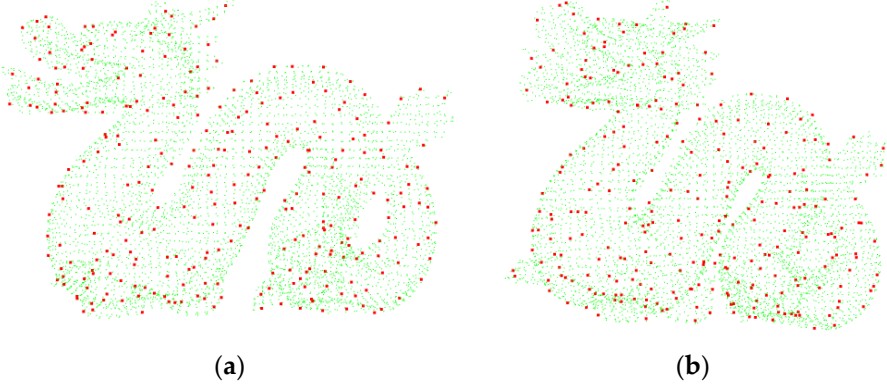

(**a**)        (**b**)

**Figure 8.** Feature points plot extracted from Dragon1 and Dragon2 models. (**a**) Dragon1 model. (**b**) Dragon2 model.

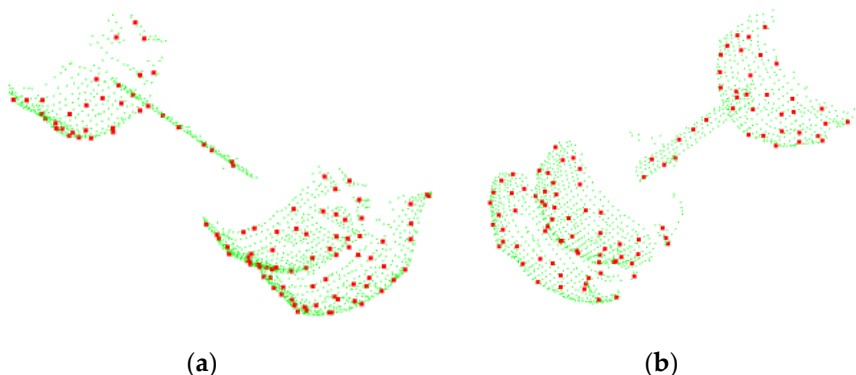

(**a**)             (**b**)

**Figure 9.** Feature points plot extracted from Drill1 and Drill2 models. (**a**) Drill1 model. (**b**) Drill2 model.

*4.2. Experimental Procedure*

In order to verify the advancement and effectiveness of the improved voxel filter, the traditional ICP algorithm, the ICP algorithm with voxel filter and the ICP algorithm with the improved voxel filter are used for registration comparison of the four models. The standard to measure the registration accuracy uses the *getFitnessScore* in PCL, which is the average of the squared distances of all corresponding points after registration. The smaller the value is, the smaller the error is, which is defined as:

$$getFitnessScore = \frac{\sum_{i=1}^{N}(p_i - q_i)^2}{N} \tag{11}$$

where $p_i$ is any point in the source point clouds $P$, $q_i$ is the corresponding point in the target point clouds $Q$, and $N$ is the corresponding point pairs after the completion of ICP fine registration iteration based on KD tree.

Table 3 shows the registration results of the three algorithms for the four models. Figure 10 shows the comparison curves of registration time and registration error of three algorithms under four model conditions. In the figure, the left axis is the registration time, and the right axis is the registration error. It can be seen that the registration time of the traditional ICP algorithm is longest, and the registration accuracy is sometimes high and sometimes low with the change of the model. For the ICP algorithm with voxel filter, the registration is faster after down-sampling, but the registration accuracy is significantly reduced. For the ICP algorithm with improved voxel filter, not only the registration time is reduced, but the registration accuracy can be stable in a higher range. The results of this registration comparison show that the improved voxel filter can more accurately represent the original data, which has stronger enhancement to the extraction of key points, coarse registration and fine registration. The more accurate the data, the faster the registration speed will be.

**Table 3.** Registration results of the three algorithms for the four models.

| Model | ICP | | ICP+ Voxel Filter | | ICP+ Improved Voxel Filter | |
|---|---|---|---|---|---|---|
| | Time | Registration Error | Time | Registration Error | Time | Registration Error |
| Bunny | 50.451 | $6.62234 \times 10^{-5}$ | 2.210 | $1.96057 \times 10^{-4}$ | 3.283 | $4.37225 \times 10^{-6}$ |
| Armadillo | 45.606 | $4.15124 \times 10^{-5}$ | 3.162 | $2.2289 \times 10^{-5}$ | 3.017 | $2.24228 \times 10^{-5}$ |
| Dragon | 40.574 | $9.28518 \times 10^{-3}$ | 6.246 | $9.12843 \times 10^{-3}$ | 6.202 | $3.91463 \times 10^{-3}$ |
| Drill | 4.558 | $1.754 \times 10^{-5}$ | 0.087 | $2.30307 \times 10^{-5}$ | 0.043 | $2.28272 \times 10^{-5}$ |

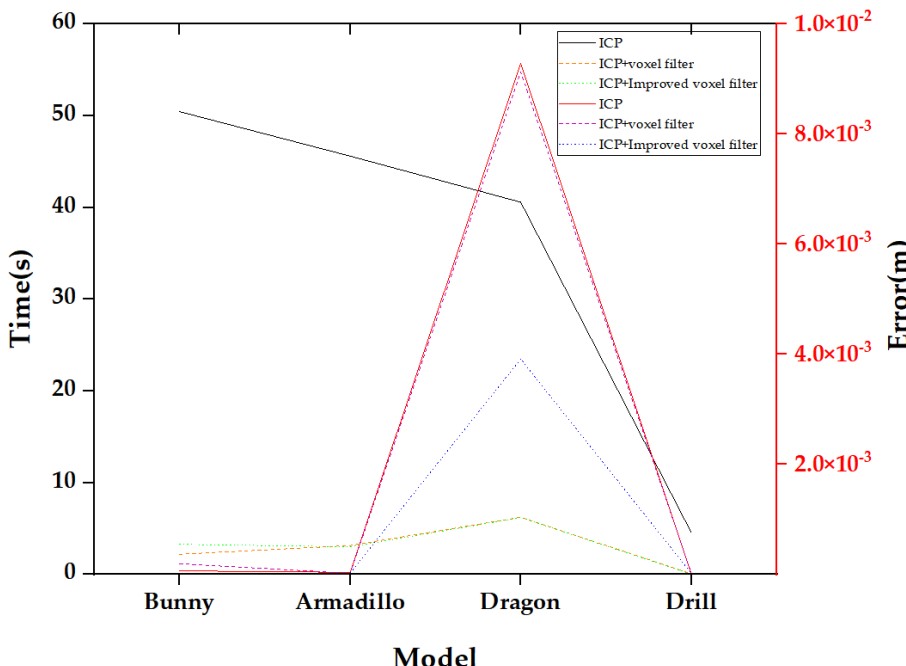

**Figure 10.** Time and Error of ICP, ICP+ voxel filter and ICP+ Improved voxel filter.

In order to verify the validity and accuracy of the proposed algorithm, three other algorithms are designed for comparative experiments under the same conditions. The three other algorithms are: the 3DSC + RANSAC + ICP algorithm without improved voxel filter, the 3DSC + RANSAC + ICP with improved voxel filter and the USC + RANSAC + ICP algorithm without improved voxel filter.

In the process of RANSAC coarse registration, adding a pre-exclusion step can immediately filter out the wrong hypothetical poses, thereby saving more time to generate more other possibly correct hypothetical poses and reducing the time of coarse registration. The distance threshold $\varepsilon_{RANSA}$ is the judgment criterion for interior point, but the selection of its value will affect the registration accuracy. For the KD tree based ICP registration, the selection of the maximum distance threshold $\varepsilon_{ICP}$ between corresponding points also has a certain affect to the registration accuracy.

According to the experimental comparison and analysis of the four models' registration, set the parameters $\varepsilon_{RANSAC} = 0.001$ and $\varepsilon_{ICP} = 0.4$ of the Bunny model, $\varepsilon_{RANSAC} = 0.01$ and $\varepsilon_{ICP} = 0.03$ of the Armadillo model, $\varepsilon_{RANSAC} = 0.3$ and $\varepsilon_{ICP} = 0.3$ of the Dragon model and $\varepsilon_{RANSAC} = 0.003$ and $\varepsilon_{ICP} = 0.009$ of the Drill model.

Table 4 shows the registration time for four models of each algorithm. Figure 11 clearly shows the trend of registration time for the four models of each algorithm. It can be seen that the registration time of proposed algorithm is the shortest. Table 5 shows the registration error for four models of each algorithm.

**Table 4.** Registration time for four models of each algorithm/s.

| Model | Voxel Filter+ 3DSC + RANSAC + ICP | Improved Voxel Filter + 3DSC + RANSAC + ICP | Voxel Filter+ USC + RANSAC + ICP | Proposed Algorithm |
|---|---|---|---|---|
| Bunny | 49.321 | 25.61 | 61.607 | 20.803 |
| Arm Adillo | 25.414 | 15.134 | 33.234 | 10.296 |
| Dragon | 45.74 | 10.302 | 4.649 | 5.78 |
| Drill | 5.654 | 4.69 | 5.131 | 4.386 |

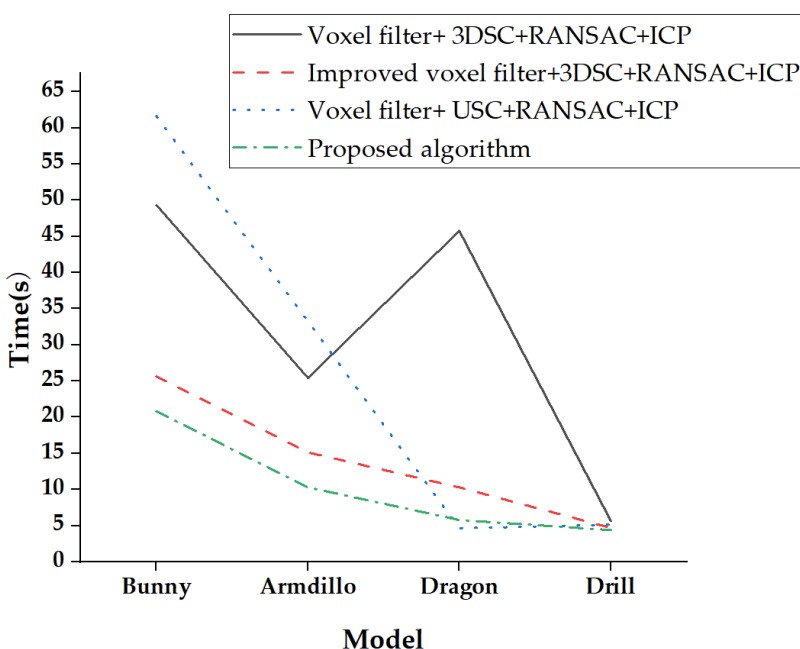

**Figure 11.** Total registration time for four models of each algorithm.

**Table 5.** Registration error for four models of each algorithm/m.

| Model | Voxel Filter+ 3DSC + RANSAC + ICP | Improved Voxel Filter + 3DSC + RANSAC + ICP | Voxel Filter+ USC + RANSAC + ICP | Proposed Algorithm |
|---|---|---|---|---|
| Bunny | $6.69635 \times 10^{-5}$ | $6.69635 \times 10^{-5}$ | $6.69635 \times 10^{-5}$ | $6.69635 \times 10^{-5}$ |
| ArmAdillo | $4.06044 \times 10^{-5}$ | $4.67503 \times 10^{-5}$ | $4.06102 \times 10^{-5}$ | $4.67117 \times 10^{-5}$ |
| Dragon | $0.00233421$ | $0.00229143$ | $0.80076627$ | $0.00228654$ |
| Drill | $4.84577 \times 10^{-6}$ | $4.68711 \times 10^{-6}$ | $4.81674 \times 10^{-6}$ | $4.71003 \times 10^{-6}$ |

For the Bunny point clouds data, the proposed algorithm is optimal in both registration error and algorithm execution time. In terms of algorithm execution time, the proposed algorithm is reduced by 50% compared with the Voxel filter + 3DSC + RANSAC + ICP algorithm, and the speed is greatly improved. Since the shape characteristics of the point clouds with the improved voxel filter are more similar to the original point clouds, the 3DSC + RANSAC + ICP algorithm combining the improved voxel filter also has a faster registration speed, and a shorter algorithm execution time.

For the Armadillo point clouds data, the Vovel filter + 3DSC + RANSAC + ICP algorithm is better than the proposed algorithm on the registration error. However, the registration error of the proposed algorithm is only $0.00058 \times 10^{-5}$ lower than the Vovel filter + 3DSC + RANSAC + ICP algorithm, with a very small difference. Compared with the Voxel filter + 3DSC + RANSAC + ICP algorithm, registration time has been reduced by 50%. When the registration accuracy error is not very large, the execution time of the algorithm can be given priority.

For the Dragon point clouds data, in terms of registration accuracy and registration time, the registration effect of the proposed algorithm is the best. Compared with the Voxel filter + 3DSC + RANSAC + ICP algorithm, the registration time of the proposed algorithm is greatly reduced. The Voxel filter + 3DSC + RANSAC + ICP algorithm with the improved voxel filter also increases the registration time by more than four times, and the registration accuracy is also higher, indicating that the improved voxel filter can improve the registration speed.

Figure 12 shows the original image of the Dragon point clouds registration, and the result under the USC descriptor algorithm without the improved voxel filter. It can be seen that when the USC descriptor algorithm lacking the improved voxel filter is used, the

registration effect of the Dragon point clouds is wrong. However, the proposed algorithm can accurately register using the improved voxel filter under the USC feature descriptor, which shows that the improved voxel filter can improve the registration accuracy. Figure 13 shows the original image of the Dragon point clouds registration and the result under the proposed algorithm.

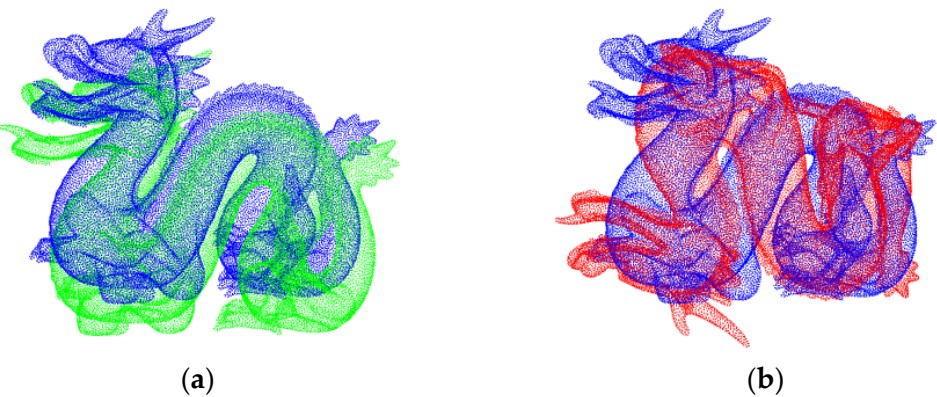

**(a)**                                                                                           **(b)**

**Figure 12.** Registration results of the USC + RANSAC + ICP algorithm without the improved voxel filter for the Dragon model. (**a**) Dragon original image. (**b**) Registration result image.

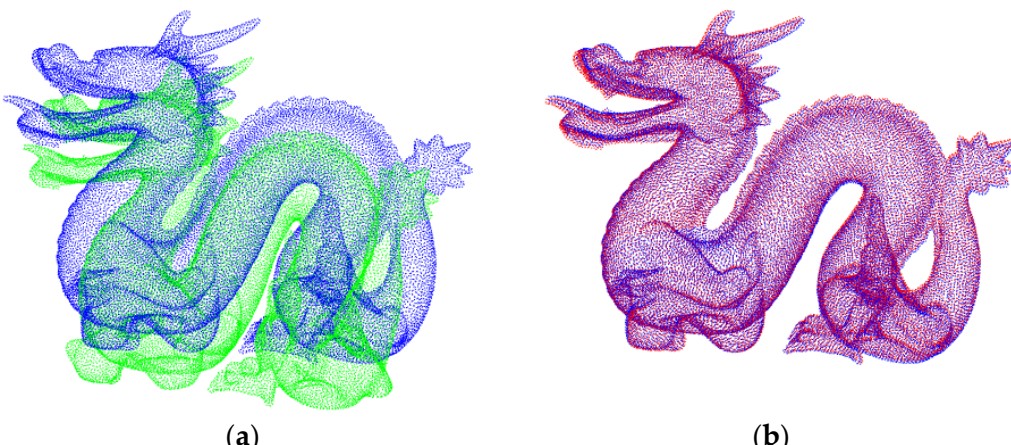

**(a)**                                                                                           **(b)**

**Figure 13.** Registration results of the proposed algorithm for the Dragon model. (**a**) Dragon original image. (**b**) Registration result image.

For the Drill point clouds data, the data is relatively simple, so there is little difference in time and registration accuracy. The registration time of the proposed algorithm is the shortest and the registration accuracy is relatively high. Whether it is 3DSC feature descriptor or USC descriptor, the registration time is reduced, and the registration accuracy is improved after using the improved voxel filter.

Figure 14 shows the registration results of four models under the proposed algorithm. The source point clouds, the target point clouds, and the registered point clouds are displayed in one figure. The source point clouds are set to green, the target point clouds are set to blue, and the registered point clouds are set to red. It can be seen that the blue point clouds and the red point clouds almost overlap, indicating that the proposed algorithm has high registration accuracy.

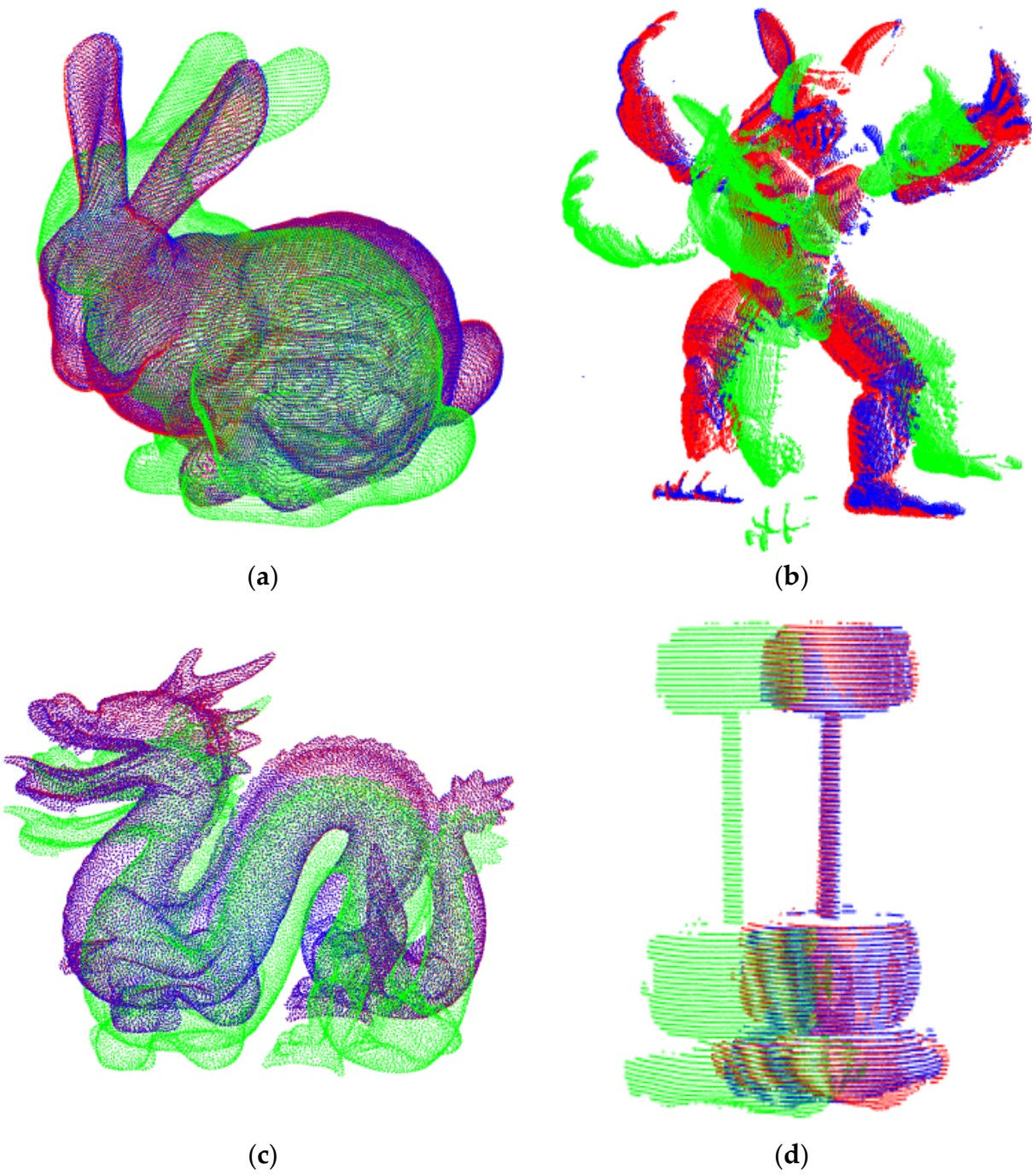

**Figure 14.** Registration results of the proposed algorithm with four models. (**a**) Bunny registration result. (**b**) Armadillo registration result. (**c**) Dragon registration result. (**d**) Drill registration result.

In order to further verify the feasibility of the proposed algorithm, the registration experiment of the VGA connector for monitor is carried out. First, a single VGA connector point clouds are obtained as the template point clouds, and then different scene point clouds are obtained as the target point clouds. The template point clouds are used as the source point clouds and the target scene point clouds for registration. The final registration pose information can be sent to the robot for decision-making. Point clouds from two different scenes are collected, and the obtained point clouds of the two scenes are shown in Figure 15.

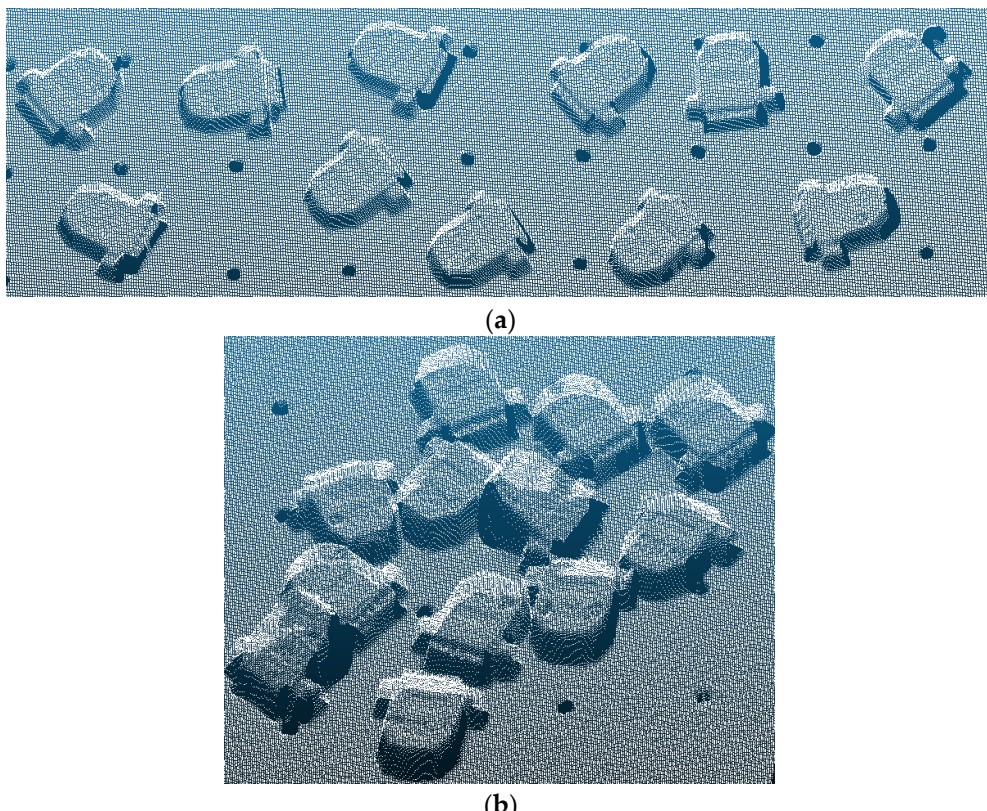

**Figure 15.** Point clouds of the two scenes. (**a**) Scene1, (**b**) Scene2.

The background of the collected scene point clouds are removed and the improved voxel filter is used to remove outliers. Table 6 shows the number of filtered points and extracted ISS feature points for the two scenes under different grid sizes. The grid size is selected by experiment to be 0.8 m. Figure 16 is the feature points plot extracted from two scenes.

An experiment is set up to register the VGA connector for monitor. Figure 17 is the operation diagram of the experimental site. Figure 18 is the registration result diagram under the two scenes. The template point clouds are set to green, the target scene point clouds are set to blue, and the registered point clouds are set to red. It can be seen that the template point clouds can be accurate with one of the target point clouds in different scenes. The position and pose of the registered point clouds obtained are sent to the robot, and the robot can stack the registered VGA connector correctly.

**Table 6.** The number of points and feature points of the two scenes under different grid sizes.

| Grid Size/m | 1 | 0.9 | 0.8 | 0.7 | 0.6 | 0.5 |
|---|---|---|---|---|---|---|
| Scene1 points | 22,356 | 25,454 | 29,576 | 34,833 | 41,461 | 49,313 |
| ISS feature points 1 | 990 | 1209 | 1473 | 1519 | 1832 | 2338 |
| Scene2 points | 19,840 | 23,057 | 27,153 | 32,386 | 39,514 | 47,423 |
| ISS feature points 2 | 1039 | 1273 | 1566 | 1538 | 1815 | 2377 |

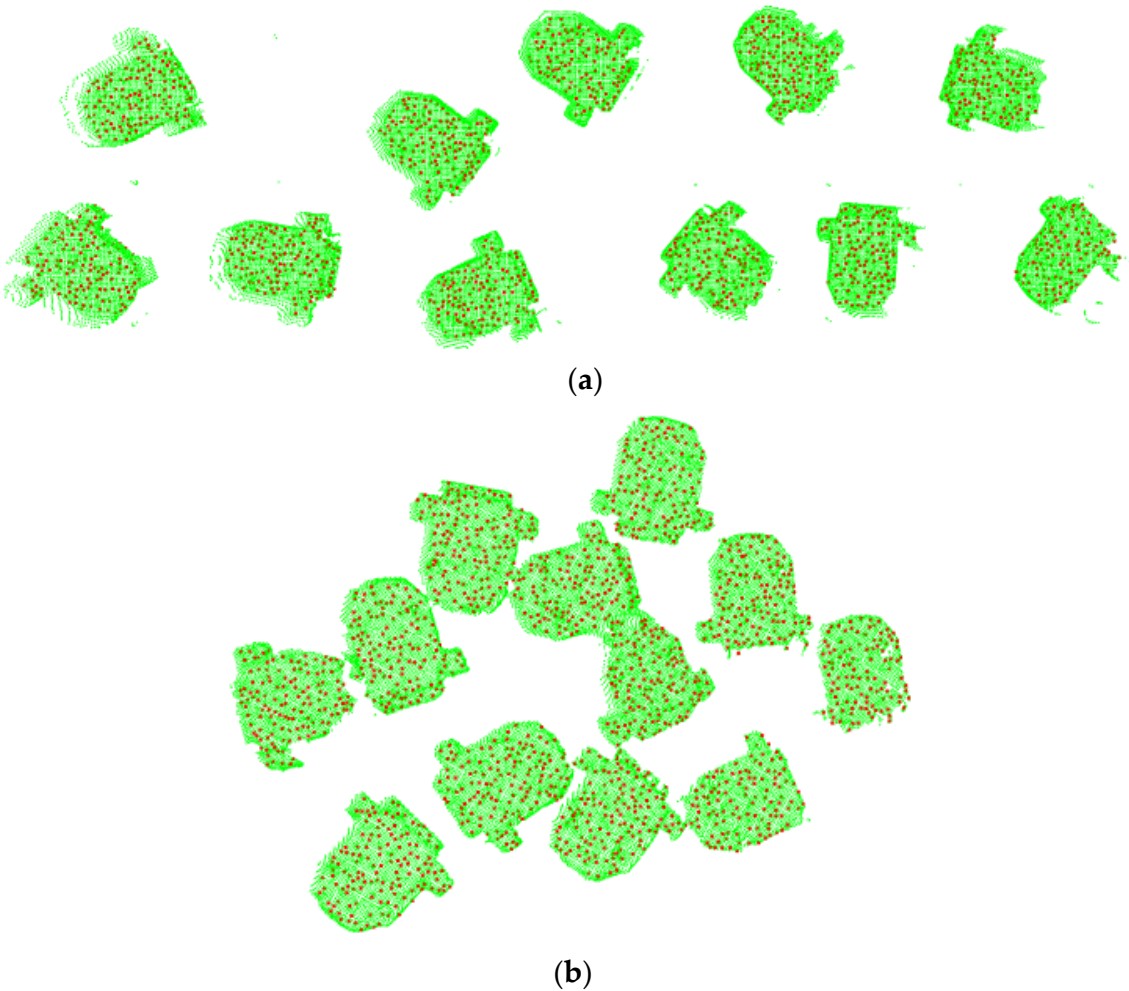

(**a**)

(**b**)

**Figure 16.** Feature points plot extracted from two scenes. (**a**) Scene1, (**b**) Scene2.

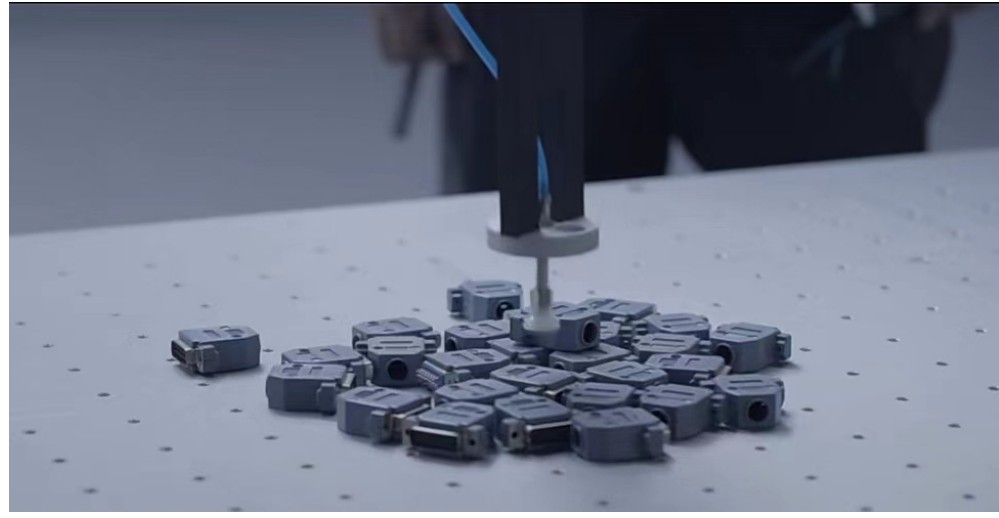

**Figure 17.** Experiment on-site operation.

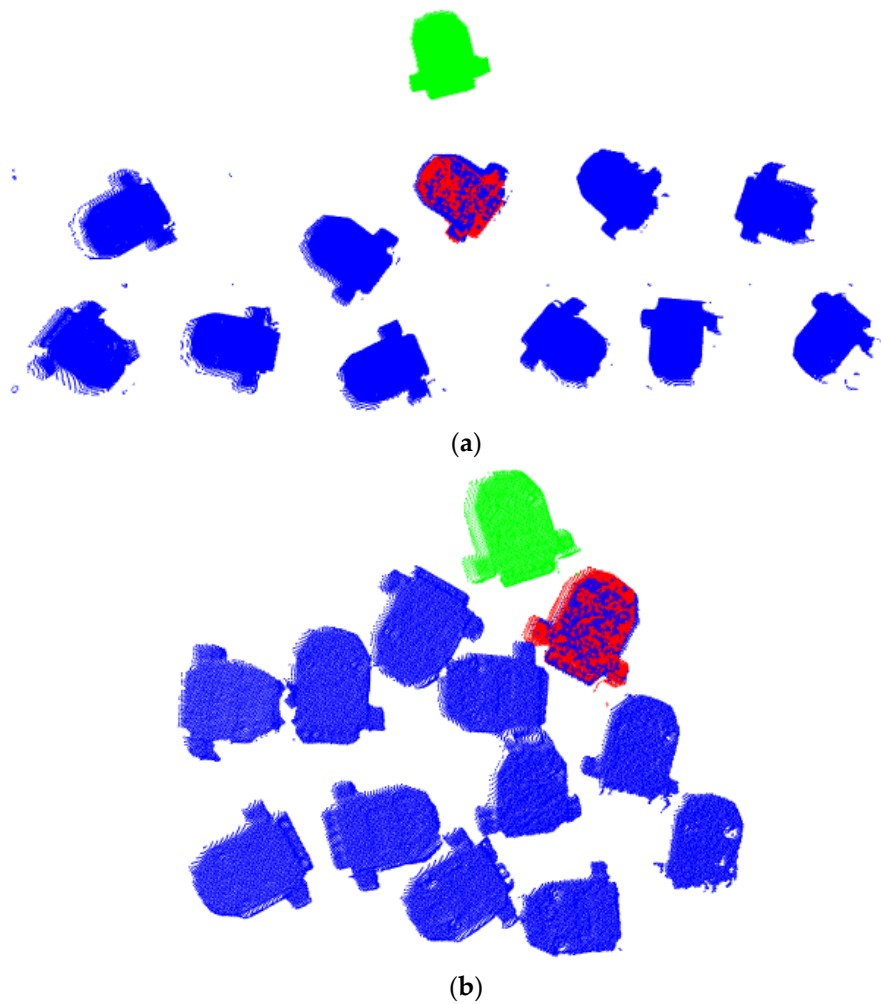

**Figure 18.** Registration results of the proposed algorithm with two scenes. (**a**) Scene1 registration result. (**b**) Scene1 registration result.

## 5. Conclusions

With the development of 3D laser scanner, point clouds have become the primary data format to represent the 3D world, and point clouds registration technology is an important step. In order to overcome the problems of slow calculation speed and low registration accuracy in the process of point clouds registration, this paper proposes a point clouds registration algorithm based on improved voxel filter combining ISS-USC features. First of all, the source point clouds and target point clouds are sampled by the improved voxel filter, respectively, to simplify the point clouds. Secondly, the ISS feature points detection algorithm is used to extract the feature points, and the USC descriptor is used to describe the feature points. Next, coarse registration is carried out by RANSAC algorithm, and the incorrect corresponding point pairs are eliminated to obtain a good initial registration position. Finally, ICP algorithm based on KD tree is used for fine registration. Through comparing with other algorithms and the registration experiment of the VGA connector for monitor, the proposed algorithm has a fast registration speed. The registration time is reduced by 50%, and the registration accuracy is higher. Considering that there are still threshold adaptation problems of some parameters in the proposed algorithm and the construction of covariance matrix that need to be further optimized, the proposed algorithm will continue to be improved in future scientific research work.

**Author Contributions:** Conceptualization, Y.D., A.W. and J.M.; software, Y.D. and A.W.; validation, A.W. and X.Z.; formal analysis, Y.D. and J.M.; investigation, A.W. and J.M.; data curation, J.M. and X.Z.; writing—original draft preparation, Y.D.; writing—review and editing, J.M. and Y.D.; supervision, A.W.; project administration, J.M.; funding acquisition, J.M. and A.W. All authors have read and agreed to the published version of the manuscript.

**Funding:** This work is financially supported in part by Natural Science Research Program of Jiangsu Colleges and Universities under Grant No. 20KJA470002, Excellent Teaching Team of "Qinglan Project" of Jiangsu Colleges and Universities, and Science and Technology Research Program of Nantong under Grant No. JC2020094 and MS22020022.

**Data Availability Statement:** The data used to support the findings of this study are available from the corresponding author upon request.

**Conflicts of Interest:** The authors declare no conflict of interest.

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
