# Peer review of "A Fast Point Clouds Registration Algorithm Based on ISS-USC Feature for the 3D Laser Scanner"

_algorithms, doi:10.3390/a15100389_

Round 1

Reviewer 1 Report (Previous Reviewer 1)

The revision does not address the most significant flaw of this work. It still lacks novelty rather than minor changes to commonly used methods. What's worse, most of these minor changes and combinations were done before already.

Author Response

Reviewer 2 Report (New Reviewer)

line 153: the phrase "... point cloud remains unchanged" is a qualitative assessment. Quantitative results on the deviation of the shape will add veracity to this claim.

line 250: explain what is meant by "expected inline score"

Equation (11) in line 330 "... and n is the logarithm on the corresponding point". What  is referred to here? Is it the number of points or something else?

Author Response

Reviewer 3 Report (New Reviewer)

In this work, the authors aim  to overcome the disadvantages of slow calculation speed and low accuracy of existing point clouds registration algorithms by proposing a fast point clouds registration algorithm based on the improved voxel filter and the intrinsic shape signature (ISS) feature point detection algorithm beside the unique shape context (USC) descriptor.

My recommendation is to accept the paper

Author Response

This manuscript is a resubmission of an earlier submission. The following is a list of the peer review reports and author responses from that submission.

Round 1

Reviewer 1 Report

The authors propose a fast point cloud registration algorithm based on the improved voxel filter and ISS-USC feature is proposed. The method follows a coarse-to-fine scheme by first running RANSAC to a rough estimate then improving with ICP to get a final result.

However, there are several fatal issues in this work. First, the overall framework is widely used [24] and [10.1109/CVPR.2015.7299195] but not proposed in this work. The claimed new  ISS-USC feature and not existing work that was used in a similar framework before [10.1109/MRA.2015.2432331, 10.1145/1877808.1877821]. The voxel filter in this work has a very minor change. The pre-exclusion in RANSAC is very common and very similar ideas have been implemented in PCL and  [10.1109/CVPR.2015.7299195].

Some experiment designs are not scientific. For example, in Table 2 line 3 and line 4, the speed is not comparing the improved voxel filter with the original voxel filter but with no filter.

Reviewer 2 Report

The author proposes a registration method that uses classical shape descriptors to establish the point-based correspondence. With a voxel-based down-sampling strategy, the point cloud can be optimized which improve the computational efficient. Combing the ISS and USC, more accurate key points can be extracted from the point cloud. Here are my comments.

1. The author declares that the improved voxel filter is more conducive to the description of the surface corresponding to the sampling point. For my understanding, it means that the proposed voxel filter achieves better geometric consistent between original point cloud and resampled one. More experimental data (Hausdorff distance) with comparisons should be reported. Some references are listed:

[1] C. Lv, Lin W, Zhao B. Approximate intrinsic voxel structure for point cloud simplification[J]. IEEE Transactions on Image Processing, 2021, 30: 7241-7255.

[2] H. Huang, D. Li, H. Zhang, et al. Consolidation of unorganized point clouds for surface reconstruction [J]. ACM Transactions on Graphics, 28:176:1–176:78, 2009.

[3] P. Mark, G. Markus, K. Leif. Efficient simplification of point-sampled surfaces [C]. In Proceedings of the conference on Visualization'02, pages 163–170. IEEE Computer Society, 2002.

2. How to handle the point cloud with different scales? Bounding box or some other solutions. The author should provide implementation details.

3. What about the performance of the proposed method for point clouds with random noisy points? Some details should be provided.

4. In current version, the test dataset is too small。 The author should report the performance in larger dataset such as ModelNet, ShapeNet or SHREC.